# Prebiotics in Global and Mexican Fish Aquaculture: A Review

**DOI:** 10.3390/ani13233607

**Published:** 2023-11-22

**Authors:** Jesús Mateo Amillano-Cisneros, María Anel Fuentes-Valencia, José Belisario Leyva-Morales, Yasser A. Davizón, Henri Marquéz-Pacheco, Gladys Valencia-Castañeda, Juan Antonio Maldonado-Coyac, Luz Adriana Ontiveros-García, Cesar Noé Badilla-Medina

**Affiliations:** 1Ingeniería en Agrotecnología, Universidad Politécnica del Mar y la Sierra (UPMYS), La Cruz 82700, Mexico; 2Ingeniería en Producción Animal, Universidad Politécnica del Mar y la Sierra (UPMYS), La Cruz 82700, Mexico; 3Centro de Investigación en Recursos Naturales y Sustentabilidad (CIRENYS), Universidad Bernardo O’Higgins, Santiago de Chile 8370993, Chile; 4Consejo Nacional de Humanidades, Ciencias y Tecnologías, Facultad de Ciencias Agropecuarias y Ambientales, Maestría en Ciencias Agropecuarias y Gestión Local, Universidad Autónoma del Estado de Guerrero, Carretera Iguala-Tuxpan km 2.5.40101, Iguala de la Independencia 39086, Mexico; 5Instituto de Ciencias del Mar y Limnología, Universidad Nacional Autónoma de México, Mazatlán 82000, Mexico; 6Facultad de Ciencias del Mar, Universidad Autónoma de Sinaloa, Mazatlán 82000, Mexico

**Keywords:** prebiotics, fish aquaculture, world, Mexico, food

## Abstract

**Simple Summary:**

The aquaculture industry is constantly growing and contributes to the food demand of the world’s population, which is already more than 8 billion people. In aquaculture, antibiotics are commonly used to mitigate the appearance of bacterial diseases that cause considerable damage to production. However, this activity induces antibiotic resistance, which risks the health of cultured organisms and consumers. Therefore, alternative supplements such as prebiotics have been emphasized, with their management and due to their multiple beneficial effects, as viable to improve the conditions of production and health of aquatic animals. In developed countries, these supplements are widespread in culture-producing commercial fish, representing a rich protein source worldwide. However, in developing countries such as Mexico, this technology is rare in commercial fish and practically non-existent in endemic fish species. The importance of such supplements for use in the small- and large-scale aquaculture of fish in developing countries should be underscored as relevant for their application in both commercial and endemic fish cultivated in developing countries.

**Abstract:**

Continued human population growth has resulted in increased demand for products, including those derived from aquaculture. The main challenge in aquaculture is producing more every year. In recent years, environmentally friendly supplements that provide the necessary pathways for optimal production have been emphasized. One of them is prebiotics, selectively utilized substrates by host microorganisms conferring a health benefit. Interest in applying prebiotics in global fish farming has increased in recent years as it has been shown to improve growth, boost the immune system, resist stress conditions, and cause the modulation of digestive enzymes. These effects reflect reduced production and disease costs. However, in Latin American countries such as Mexico, large-scale use of these food supplements is needed as a sustainable alternative to improve fish production. This paper gives a review of the current advances obtained with the application of prebiotics in commercially farmed fish worldwide, mentions the prebiotics to use in the aquaculture industry, and updates the status of studies about the used prebiotics in global commercial fish cultivated in Mexico, as well as freshwater and marine endemic fish in this country. Also, the limitations of prebiotics application in terms of their use and legislation are analyzed.

## 1. Introduction

Aquaculture activity is considered the fastest-growing industry in terms of food production worldwide. Compared to fishing, it has had an upward growth since the 1980s, and in 2020, represented the world production record with 122.6 million tons, of which 57.5 million tons belonged to fish aquaculture [1].

Since its inception, the main objective of aquaculture has been to increase production and improve profitability [2]. To this end, a high-density culture is often chosen. However, this practice, in addition to causing damage to the environment, contributes to an increase in the stress conditions of the cultivated aquatic organisms, which leads to the appearance and spread of diseases, resulting in massive mortalities with effects directly related to severe economic losses [3,4]. Traditionally, antibiotics have been widely applied for disease control and growth promoters in aquatic organisms [5]. Because of their beneficial effects, some antibiotics are approved for use in aquaculture production [6]. In recent years, it has been suggested that their use be more strictly controlled, and some antibiotics have even been banned, as their indiscriminate use can lead to the spread of antimicrobial residues in aquatic environments, increasing antibiotic resistance rates in aquatic bacteria and impacting public health because antimicrobial resistance can be transferred to pathogens and could damage human consumers [7]. Therefore, other sustainable alternatives should be investigated, such as administering prebiotic, probiotic, and synbiotic supplements [8].

Prebiotics are foods that cannot be digested or assimilated directly by the host organism [9] or by probiotics, which the FAO and WHO have defined as live microorganisms that, in adequate doses (10^6^–10^7^ CFU/g), have beneficial effects on organisms that consume them [10]. 

In developed countries, applying prebiotic supplements is common in farmed fish of global commercial importance, contributing to about 20 kg of the world’s average annual per capita consumption of aquatic food [1]. However, this consumption in Latin America and the Caribbean is only 9.9 kg per person per year [1].

Fish aquaculture in Mexico continuously grows due to the demand for nutrient-rich products (high in protein and other nutrients such as vitamins, minerals, and beneficial fats such as omega 3 and 6). However, prebiotic supplements in the Mexican aquaculture industry must be applied to high-scale production. In addition, the sources and types of prebiotics potentially usable in Mexico’s productive sector need to be discovered. Therefore, this document defines the types of prebiotics most used in global fish aquaculture and analyzes the current use of prebiotics in fish farming in Mexico. The different limitations of prebiotics application in terms of their use and legislation are analyzed. Furthermore, it is necessary to mention the new potential prebiotics that could be applied to fish species that represent world production and to the freshwater and marine endemic fish in Mexico that have aquaculture potential.

## 2. Importance and Types of Prebiotics 

Although the host hardly assimilates prebiotics, they are the basis of a highly beneficial metabolic cascade for the host since they serve as a fundamental substrate to be fermented by bacteria with beneficial characteristics such as *Bifidobacterium* and *Lactobacillus*, bacteria genera found in the digestive system of the host organisms. These genera have been found to have metabolic faculties related to forming short-chain fatty acids (acetic, propionic, and butyric acids), decreasing the gastrointestinal pH [11,12,13,14,15].

This type of metabolites can be absorbed and used as a source of energy. In addition, in the host organism, they generate various benefits such as the reduction or inhibition of pathogenic bacteria by modulating the microbiota [12,13,16], increased health of the digestive tract and the whole organism [9,17], along with the increased availability of micronutrients such as calcium, magnesium, iron, and sodium [13,18,19].

Prebiotics can be found in many foods such as tomatoes, bananas, apples, oranges, grapefruit, papaya, mango, garlic, onion, broccoli, corn, potato, peas, lentils, beans, chickpeas, oat, barley, among others [20]. Hendry [21] indicated that about 36,000 plant species store carbohydrate fractions as the prebiotics inulin or fructooligosaccharides.

Over the years, prebiotics have been classified from non-digestible carbohydrates, which include arabinoxylan-oligosaccharides (AXOS), β-glucans, stachyose, fructo-oligosaccharides (FOS), galacto-oligosaccharides (GOS), inulin, isomaltooligosaccharides (IMO), lactylol, lactosucrose, lactulose, mannan-oligosaccharides (MOS), oligofructose, transgalactooligosaccharides, and xylooligosaccharides (XOS) [12,22]. However, prebiotics differ from carbohydrates, peptides, proteins, and lipids [20,23]. This list of prebiotic supplements is continuously growing due to the diversity of organisms that can be used to obtain them.

## 3. Feasibility and Future Demand for Prebiotics in the Global Industry

From the efficiency point of view in terms of time and maintenance expense, the addition of prebiotics does not require remarkable production technologies (e.g., viability testing during processing and storage as applied in probiotic use), as these ingredients are not altered by environmental conditions such as air and heat [24].

The demand for prebiotic supplements has continually increased worldwide over the years [25]. The demand for prebiotics is expected to reach 1.35 million tons by 2024 [26]. In terms of the price of this supplement on the global market, the Market Analysis Report indicates that by 2021, the market for prebiotics was valued at over $6 billion and is expected to grow by 14.9% annually from 2022 to 2030 [25].

From the classification of different types of prebiotics, inulin dominated the global market in 2021 and accounted for 37% of prebiotic production, and a growing demand for this supplement as part of beverages and baked goods is presumed. In addition, it has been defined that the global demand for GOS will grow significantly in the next seven years [25].

In Mexico, there needs to be more information about reliable price numbers and the costs of commercial use of these supplements. Therefore, a thorough analysis of their commercial uses and prices is required.

## 4. Characteristics of Prebiotics

In the area of human nutrition, for a prebiotic to be considered as such, it must meet the following characteristics described by Gibson et al. [27]: (1) resist gastric acidity, hydrolysis (mammalian enzymes), and absorption in the anterior digestive system, (2) be fermented by the intestinal microbiota, and (3) selectively stimulate the growth of intestinal bacteria related to the health of the host organism. Among these three points, the third is the most difficult to achieve and, therefore, the most important because determining it requires anaerobic sampling and viability testing of various bacterial species [27].

### 4.1. Obligatory Characteristics of Prebiotics in Aquaculture

The characteristics of prebiotics used in human nutrition are different from aquaculture. According to Lauzon et al. [13], the classification characteristics that prebiotics have in human nutrition are not sufficient for aquaculture since, through this activity, species of organisms of various taxonomic groups (e.g., algae, mollusks, crustaceans, fish) are used, as it is evident that they have different energetic requirements and capacity to consume these supplements. For example, the structure of the gastrointestinal tract of fish shows significant variation between species [13], as there are more than 25,000 fish species [28]. Fish with different feeding habits (carnivorous, herbivorous, and omnivorous) are cultured, and there are fish with and without stomachs, making it more complicated to define the role of prebiotics. Given these variations in aquatic animals, Lauzon et al. [13] suggest that prebiotics should: (1) resist hydrolysis by the host organism enzymes and resist absorption by the gastrointestinal tract, (2) improve the microbial balance of the intestinal tract, and (3) be beneficial to the host (e.g., improved disease resistance, increased survival, and non-specific immunity, modulation of the intestinal morphology, as well as increased nutrient digestibility and growth). The third point on this list is focused on the beneficial results of studies where different prebiotic types have been applied to various fish species, where these supplements influence various mechanisms directly related to improving growing conditions (Figure 1). However, prebiotics previously applied with favorable results in some species, when added to other species by changing the dose and application times, obtained negative results or were without effects on some fundamental mechanisms that are attributed to improve aquaculture production (Table 1). 

### 4.2. Mechanisms of Prebiotics

It has been defined in multiple studies that prebiotic supplements, when added mainly to the diet of fish in juvenile stages and controlled cultivation, interact with different mechanisms, which are directly related to the improvement of growing conditions in terms of the growth, health, and survival of organisms (Table 1).

The mechanisms influenced by prebiotics are an increased production of digestive enzyme and short-chain fatty acids, number of enterocytes, and size of intestinal microvilli; improvement of the immune system with increased blood cells and gene expression; modulation of the presence of intestinal microorganisms increasing beneficial bacteria such as lactic acid bacteria and reduction in pathogenic bacteria; and, in addition, reduction in the presence of cortisol and triglycerides (Figure 1).

## 5. Use of Prebiotics in Global Fish Aquaculture

In recent years, there has been a growing interest in determining the effects produced by prebiotic supplements in global fish aquaculture. The main prebiotics used include inulin [47,49,50,51,52,53,56,57,58,61], β-glucans [30,31,32,34,36,37,40,69], FOS [50,61,70], MOS [52,61,65,69,70], GOS [43,61,69,70,71], XOS [68,70], AXOS [29], IMO [45], and GroBiotic^®^-A (International Ingredient Corporation, Lake City, MN, USA) [69] (Table 1).

Studies have defined the different effects caused by the application of prebiotics in fish culture of economic and commercial interest, such as catfish, carp, salmonids, and tilapia, the organisms that have represented the highest percentages of world production in inland, coastal, and marine aquaculture for the last 20 years [1]. These fish belong to the families Cichlidae, Clariidae, Cyprinidae, Pangasiidae, and Salmonidae (Table 2).

As for the types of prebiotics added to fish diets, trends regarding their multiple effects are identified.

I.In studies where β-glucan is added to different fish species, the results are mainly in improving growth and immune-related mechanisms such as phagocytosis, lysozyme, haemolytic complement, and bactericidal activity (Table 1).II.Regarding the application of inulin, in different studies, the effects are presented in an increase in the production of digestive enzymes, and short-chain fatty acids, as well as improvement of the immune system and growth. In contrast, in other studies, there are negative and null results in growth, survival, and the intestinal cells (Table 1).III.MOS has been found to have multiple beneficial effects such as improved growth, survival, immune system, length and density of microvilli of intestinal cells, as well as a reduction in potential intestinal pathogenic bacteria such as *Aeromonas* spp. and *Vibrio* spp. Similar results are found with the cell wall application of *Saccharomyces cerevisiae*, which is composed of β-glucan and MOS (Table 1).IV.Works where AXOS, GOS, oligofructose, and XOS are applied are less abundant compared to works studying β-glucan, inulin, and MOS. Despite this, the results have been beneficial for growth, survival, and the immune system, as well as an increased activity of the digestive enzymes (lipase and amylase) and microvilli length of the intestinal cells (Table 1).

Based on the analysis of the effects of prebiotics on fish species, the importance of their application in improving aquaculture conditions is emphasized.

Research worldwide shows that the benefits of prebiotics supplements include an improved immune response and nutrient absorption (increased enterocyte number and microvilli size), as well as increased growth and survival rates (Table 1) [30,31,32,35,40,42,44,49,51,52,53,54,55,59,60,64,65]. Mo et al. [52] studied in the grass carp *C. carpio* the effect of inulin and mannan-oligosaccharides at concentrations of 0.2 and 2% in the administered diet. After 56 days of feeding, both prebiotics increased growth. In another work, Abu-Elala et al. [40] found that applying *S. cerevisiae* cell walls composed of β-glucans and mannan-oligosaccharides at 0.1 and 0.2% in the tilapia *O. niloticus* for nine weeks improved growth and the immune system. In another study, Ren et al. [65] applied MOS at different concentrations (0.3, 0.6, 1.0, and 2.0%) in the grouper *Epinephelus* after 9 weeks and found that they had an increase in lysozyme activity related to the increase of the immune system, as well as an increase in the size of the microvilli. Zhu et al. [42] found in hybrid grouper *Epinephelus* that MOS and XOS improved growth and survival by 28 days at concentrations of 0.2 and 0.05%, respectively.

Not all specific factors involved in the effectiveness of prebiotics in fish are known with certainty. Some model studies have tried to explain extensively the mechanisms related to the effects of prebiotics on fish, principally in the gut microbiota and their relationship to factors that benefit production, such as growth and survival. For example, in the work reported by Geraylou et al. [29], applying 2% arabinoxylanoligosaccharide to the diet of the Siberian sturgeon *A. baerii* for 12 weeks increased the number of lactic acid bacteria such as *Lactobacillus*, *Lactococcus*, and *Clostridium*, as well as the presence of short-chain fatty acids, which are considered precursors of growth-related pathways, so it was concluded that via these pathways and with the presence of prebiotic AXOS, there was a significant improvement in growth. Tiengtam et al. [53], when adding inulin as a supplement to *O. niloticus*, found that there was an increase in the number of enterocytes as well as an increase in the size of the microvilli of these cells. With these characteristics, this species can assimilate more nutrients and thus improve their growth. While Akter et al. [64], by applying mannanoligosaccharide to the catfish *P. hypophthalmus* at different concentrations (0.2, 0.4, 0.6, and 0.8%) for 12 weeks, increased survival by decreasing the pathogenic bacterium *A. hydrophila*., the better survival of which may reflect a higher production capacity of aquaculture activity.

However, in a few studies, adverse effects resulting from the application of prebiotics in fish have also been seen, ranging from decreased growth performance [22] and enzyme activity in the immune system [61] to gut-related problems such as decreased numbers of enterocytes [46], decreased villi heights [72], and intestinal inflammation [73] (Table 1).

## 6. Limitations of Use of Prebiotics in Fish Aquaculture

The results concerning the effects caused by the application of prebiotics in fish aquaculture have been very variable in different studies in recent years. These results vary according to the fish species to which these supplements are added, the age of organisms, their diet (carnivore, herbivore, omnivore), the environment where they develop (cold, warm, freshwater, marine), the type or types of prebiotics applied, doses and period of application, as well as the environmental conditions (physical/chemical factors) where the experiments are carried out. In addition, it is crucial to know the nutritional requirements of each fish species to which prebiotics are added since an inadequate dose can be potentially dangerous or have no effect on the organisms (Table 1). Therefore, before applying any prebiotic to aquaculture production, it is necessary to evaluate dose/time tests with model organisms, to define their behavior, if a specific prebiotic has not been previously applied to them. Also, the use of prebiotics in species with physiological characteristics like those that have previously had favorable results can be considered.

Another limitation to be considered when applying prebiotics is the regulations. There are no regulations for including prebiotics in aquaculture feed, or the existing ones are minimal. The only one that exists is for application in humans, which is different among the countries that use it [74,75]. The scientific evidence on the effectiveness of these ingredients is agreed upon by Japan, the European Union, the United States of America, Argentina, and Brazil; likewise, these nations add prebiotics and probiotics similarly as if they were the same [75]. Although both are closely related regarding their benefits they can have on the organism that consumes them, they have essential differences. Probiotics must be live microorganisms that are applied in specific amounts (between 10^6^ and 10^9^ CFU/g, 10) and this quantity must be carefully established before their application as a probiotic. At the same time, prebiotics are non-living ingredients and the care in applying them is not as critical as with probiotics. Concerning these characteristics that differentiate prebiotics and probiotics, it is appropriate to apply specific legislation to each supplement for its handling and application to the diets of humans and aquatic organisms separately to gain control over its use and its effects. In addition, these laws must be homogeneous for developed and developing countries due to the globalization of product markets, thus contributing to more efficient resource management.

Furthermore, it is noteworthy that in different studies, high-tech tools have already been applied both for the analysis of metabolites modulated by prebiotics [29,76] and for gene analysis (DNA and RNA) on the influence of these supplements [40,43,66,77]. However, it would be interesting to conduct studies using other “omic” technologies simultaneously to obtain broader schemes of the effects of these supplements that can serve as a guide to be applied in improving aquaculture production levels.

## 7. Potential New Prebiotics in the Fish Aquaculture Industry

Despite the limitations of the use of prebiotics, there is continuous research on the analysis of potential new prebiotics. For this purpose, organisms of various taxa such as algae, fungi, invertebrates such as crustaceans and insects, as well as fish rich in these supplements are used. One of them is chitosan, found in fungal cell walls, annelid chitin, some crustacean exoskeletons [78], and fish scales [79]. The latter is quite interesting regarding the raw material quantity used for this supplement. For example, the global tilapia production for 2020 represented 11.2 percent of the total for inland aquaculture (Table 2).

Chitosan is a potential prebiotic due to its non-toxic, antimicrobial, antioxidant, biocompatible, and biodegradable properties. Due to its diverse properties, it has been used in seafood preservation [80], fruits, and minimally processed vegetables [81,82]. This potential prebiotic has also been used in some works related to fish aquaculture and has been found to have different beneficial effects, such as those established in the required characteristics of prebiotics [13]. Despite the characteristics in terms of the effects on the improvement of the cultivation conditions found with chitosan, it is necessary to check its beneficial effects in more experiments carried out in different species of fish to have more certainty about its application.

Geng et al. [83] noted increased growth and immunity against the pathogenic bacterium *V. harveyi* by adding doses of 0.3 and 0.6% chitosan to the diet of the cobia fish *Rachycentron canadum*. The review by Ahmed et al. [84] indicated that chitosan added to diets significantly improves growth and survival in both freshwater (*C. carpio*, *Carassius gibelio*, *Cirrhinus mrigala*, and *O. mykiss*) and marine (*Epinephelus bruneus*, *L. calcarifer*, *Paralichthys olivaceus*, and *Scophthalmus maximus*) fish. Meanwhile, Yildirim-Aksoy and Beck [85] indicate that chitosan significantly reduces the presence of highly pathogenic bacteria (*A. hydrophila*, *Edwardsiella ictaluri*, and *F. columnare*) from warm water finfish (catfish, carp, and tilapia).

Other potential prebiotics are laminarian and fucoidan, which are extracted from the brown alga *Sargassum* [86], which has wreaked havoc in recent years on Mexican Caribbean beaches. El-Boshy et al. [87], when applying laminarian in the tilapia *Oreochromis niloticus*, found increased bactericidal activity (lysozyme) and survival against the pathogenic bacterium *A. hydrophila*. In another study, Immanuel et al. [88], when applying fucoidan in *Penaeus monodon* shrimp, found that it increased innate immunity and survival against the white spot syndrome virus, which has caused multi-million-dollar losses in the shrimp industry since it appeared in early 1990. In a review by Raposo et al. [89], they analyzed different prebiotics, such as alginate, laminarian, and fucoidan, extracted from macro and microalgae (genera *Ascophyllum*, *Fucus*, *Laminaria*, *Sargassum*, and *Undaria*), and found that most of these ingredients have effects in an increase in beneficial bacteria (*Bifidobacterium*, *Lactobacillus*), an increase in short-chain fatty acids, and a reduction in pH in mammals such as mice, pigs, and humans. Additionally, Allsopp et al. [90] conducted an in vitro investigation on the potential of the xylan prebiotic extracted from the red alga *Palmaria palmata* and found that this ingredient increased the production of short-chain fatty acids acetate, propionate, and butyrate. The authors proposed further investigating these prebiotics added to human diets based on this result. As for applying this diversity of prebiotics found in the algae mentioned, it is necessary to design experiments where they add prebiotics together (consortium) isolated from these organisms to define their potentially synergistic effects on the growing conditions of aquatic organisms. The use of algae as supplements in developing countries is undervalued due to ignorance about their potential applications.

Given the cited works in this section, it is evident that there is still much to be tested in the research on prebiotics being applied in fish aquaculture. In addition, it is necessary to inform and convince the people within the aquaculture sector of the benefits of these potential prebiotics to apply them since they are in the habit of continuously applying what works, and are not updated on the application of new technologies that could improve their productive activity.

## 8. Current Status and Potential Use of Prebiotics in Mexican Fish Aquaculture

### 8.1. Fish Production in Mexico

In Mexico, fish production represents second place in terms of cultivated aquatic organisms, followed by crustacean farming [91]. Fish aquaculture in Mexico has fluctuated with an increasing trend in recent years [91]. However, with the COVID-19 pandemic, this trend was interrupted by a significant reduction in the production of fish (Figure 2). The pandemic of COVID-19 has caused since, the first months of the year 2020, various damages to health, society, and economies, as well as fishing and aquaculture activity, in many countries of the world. Fishing and aquaculture were affected by a reduction in the growth trend over the last two decades [1]. In addition, the confinement and closure of markets caused by this disease in 2020 affected international trade in fishery and aquaculture products, generating 151 billion USD, compared to the 165 billion USD recorded in 2018 [1].

In terms of the production quantity presented in Mexico, the main fish species farmed are the tilapia *Oreochromis niloticus*, two species of carp (the common carp *C. carpio* and grass carp *C. idella*), the rainbow trout *O. mykiss*, and the channel catfish *Ictalurus punctatus* [91] (Figure 2).

### 8.2. Use of Prebiotics in Mexican Fish Aquaculture Experiments

Although there have been numerous studies worldwide where prebiotics have been applied to the commercial fish species that are among the most cultivated in Mexico, it is appropriate to mention some of those carried out in this country since the climatic conditions may be different in other countries, which may cause changes in the effects previously established, even for the same species. For example, when applying FOS to rainbow trout with a concentration of 0.5% for 70 days, Cid-García et al. [92] found no effect on growth. In contrast, there was a significant increase in lipid and protein content in the muscle and the immune system (Table 3). In another study using the same fish species, Segura-Campos et al. [93] found no growth changes for both prebiotics when adding FOS and MOS to 3% of their diet for 60 days. However, MOS increased the lipid content in the muscle (Table 3).

In the Nile tilapia *O. niloticus*, Flores-Méndez et al. [94] applied the experimental prebiotic agavina derived from *A. tequilana* in concentrations of 2% and 4% for 80 days in normal conditions and 30 days in crowded conditions, where at 80 days, there was no effect on growth. While under space stress conditions, they found an improvement in growth and a significant reduction in cortisol stress hormone and triglyceride levels (Table 3).

In addition to these studies, in recent years, different prebiotics have already been applied in freshwater and marine endemic fish from Mexico and neighboring countries (e.g., channel catfish, Pacific red snapper, tropical gar, leopard grouper, and totoaba), where the effects vary by species, the dose, time of application, and the type of these supplements.

In the freshwater fish channel catfish *Ictalurus punctatus*, Sánchez-Martínez et al. [95] noted that a concentration of β-glucan of 0.05% for 28 days increased the immune system. In contrast, Guzmán-Villanueva et al. [96] added the same prebiotic with concentrations of 0.1 and 0.2% to the marine fish Pacific red snapper *L. peru* for 42 days and found that it had significant effects on improvement in growth and the activity of the digestive enzymes trypsin and chymotrypsin.

In the freshwater fish tropical gar *A. tropicus*, prebiotics have been applied where the results are variables with positive, negative, and null effects in the analysis of growth and activity of digestive enzymes (Table 3). Nieves-Rodríguez et al. [97] and Cigarroa-Ruiz et al. [98], when adding β-glucan in doses of 0.2–2.0%, by 62 and 21 days, respectively, saw no effects on growth. In contrast, the effects on digestive enzymes were different between these studies. In the first study, the effects were null, and in the second, there was a significant increase in trypsin and lipase activities.

In studies using FOS in concentrations of 0.5–2.0%, Sepúlveda-Quiroz et al. [99] noted increased growth and a significant reduction in protease, trypsin, and lipase activities. While Pérez-Jiménez et al. [100] found significant increases in protease and amylase activities. Regarding the use of inulin, De La Cruz-Marín et al. [101] defined an increase in survival at a concentration of 2.5% after 45 days of the experiment. However, they found a negative effect on growth at 1.0 and 1.5% concentrations. In another study, Maytorena-Verdugo et al. [102], when adding MOS in different concentrations (0.2, 0.4, and 0.6%) for 20 days, found a significant increase in growth and digestive enzymes trypsin, lipase, and amylase.

**Table 3 animals-13-03607-t003:** Summary of research in Mexico on the effects of prebiotics supplements in fish aquaculture.

Prebiotics	Fish Species	Dosage/ Application Time	Prebiotics Sources	Effects	References
Agavin	*Totoaba macdonaldi*	2%/44 days	*Agave tequilana*	→ Growth	[101]
Agavin	*O. niloticus*	2, 4%/80 and 110 days	*A. tequilana*	80 days → growth 110 days ↑ growth, ↓ cortisol and triglycerides level	[91]
Agavin	*T. macdonaldi*	1%/56 days	*A. tequilana*	→ Growth, trypsin and protease activity	[102]
β-glucan	*Lutjanus peru*	0.1, 0.2%/42 days	*S. cerevisiae*	↑ Growth, trypsin, and chymotrypsin activity	[93]
β-glucan	*I. punctatus*	0.05, 0.1, 0.5%/28 days	*S. cerevisiae*	0.05% ↑ immune system	[92]
β-glucan	*Atractosteus tropicus*	0.5, 1.0, 1.5, 2.0%/62 days	*S. cerevisiae*	→ Growthand protease, trypsin, and amylase activity	[94]
β-glucan	*A. tropicus*	0.2, 0.4, 0.6, 0.8%%/21 days	*S. cerevisiae*	→ Growth 0.6 and 0.8% ↑ trypsin and lipase activity	[95]
β-glucan, chitosan and inulin	*T. macdonaldi*	0.1, 0.5, 1.0%/60 days	β-glucan from *S. cerevisiae* Chitosan from shrimp shells Inulin from *Agave* sp.	β-glucan ↑ gene immune system Chitosan ↑ respiratory burst ↓ Lipase activity Inulin ↑ trypsin and lipase activity	[103]
FOS	*O. mykiss*	0.5%/70 days	*Saccharum officinarum*	→ Growth ↑ Immune system and lipid and protein in muscle	[89]
FOS	*A. tropicus*	0.5, 1.0, 1.5, 2.0%/45 days	*A. tequilana*	↑ Growth ↓ Protease, trypsin, and lipase activity	[96]
FOS	*A. tropicus*	0.5, 0.75%/15 days	*A. tequilana*	0.5% ↑ protease and amylase activity 0.75% ↑ growth, survival	[97]
FOS y MOS	*O. mykiss*	3.0%/60 days	FOS from *S. officinarum* MOS from *S. cerevisiae*	→ Growth, protein in muscle MOS ↑ lipid in muscle	[90]
Inulin	*Mycteroperca rosacea*	1%/56 days	*A. tequilana*	→ Growth and lisozyme activity	[100]
Inulin	*A. tropicus*	0.5, 1.0, 1.5, 2.0, 2.5%/45 days	*A. tequilana*	1.0 and 1.5% ↓ growth 2.5% ↑ survival	[98]
MOS	*A. tropicus*	0.2, 0.4 and 0.6%/20 days	*S. cerevisiae*	↑ Growth and trypsin, lipase, and amylase activity	[99]

Symbols represent increase (↑), no effect (→), or decrease (↓) in the response parameter of the prebiotics relative to the control.

In another work, Reyes-Becerril et al. [103], when adding inulin at 1% in the leopard grouper *M. rosacea* for 56 days, there were no changes in growth and lysozyme activity. The same result was obtained in the Totoaba *T. macdonaldi* using the experimental prebiotic agavin in concentrations of 1 and 2% [104,105]. In the same species, the effects of β-glucan and inulin were noted with increased genes in the immune system and digestive enzyme activities, respectively [106].

The good results obtained in research conducted in different parts of the world regarding the application of prebiotics in farming fish species including tilapia, carp, rainbow trout, and different catfish species, as well as the variable results of species studied in Mexico, should be considered in order to evaluate the effects of different prebiotics, with variations in doses and application times, in endemic fish with great aquaculture potential in Mexico, such as those belonging to the families Atherinopsidae, Cichlidae, Cyprinidae, Ictaluridae, Lepisosteidae, Lutjanidae, Salmonidae, and Sciaenidae. It should be noted that the application of prebiotics in experiments conducted in Mexico has increased in recent years. However, it is necessary to share the knowledge generated with small and large aquaculture producers, who, with an adequate application of these supplements in the diet of the organisms used, can contribute to an increase in fish aquaculture in Mexico, generating more economic profit and necessary jobs throughout the process of production and sale of products in this activity.

In addition, in studies analyzing the effects of prebiotics on fish, a continuous application of high-tech tools related to omic sciences is necessary, which can help us understand and develop a better explanation of what happens in aquatic organisms to which these food supplements are applied.

## 9. Challenges and Opportunities

Research shows that prebiotics have multiple beneficial effects in improving fish farming conditions, reflecting increased growth, health, and survival. Despite this, different aspects limit their optimal use in productive activity in developing countries. So, there are some challenges and opportunities, which include:I.The basis of known and applied prebiotics with beneficial effects in fish aquaculture with global commercial importance results in an opportunity to advise small, medium, and large producers in developing countries on their application and improvement of their activity.II.To make known the alternatives of new potential prebiotics to experiment and test the effects on commercially known and endemic fish that can be used for production.III.To consider the opportunity of taking advantage of undervalued natural sources of prebiotics, such as the large amount of *Sargassum* reaching the coasts of Mexico and the waste of organisms, such as crustaceans, potential sources of chitosan, to isolate these supplements and design experiments to determine their effects on the global and endemic commercial fish used in aquaculture from developing countries such as Mexico.IV.To experiment with the addition of prebiotics and consortia of prebiotics by varying their concentrations, time, and dose of application to determine whether there is synergy in the benefits.V.Make more timely legislation in developing countries on the use of prebiotics in the aquaculture sector, considering the laws applied in developed countries in this productive activity.

## 10. Conclusions

Prebiotics stand out for positively modifying the presence of beneficial microbiota, whose function is the modulation of different metabolic pathways for the adequate maintenance of the organisms that consume these food supplements, and their results are reflected in the improvement of the culture conditions in terms of the growth, modulation of digestive enzymes, and health (improve the immune system) of aquatic organisms. However, multidisciplinary studies such as molecular analysis using different tools (e.g., DNA and RNA sequencing, and metabolites analysis) are needed to explain these supplements’ effects better.

In aquaculture, despite positive results with the application of prebiotics, there are variations in the effects in the species to which they are added, age, dosage, period of application, and environmental conditions. Extensive studies are needed to consider the nutritional requirements of each species and in different culture conditions, which can be influenced by physical/chemical factors (e.g., temperature, pH, dissolved oxygen, nitrogen) or stressors such as overcrowding. Also, it is desirable to examine in more detail the potential prebiotics that can be used in the aquaculture sector and that are already applied in other areas, such as human and veterinary medicine for terrestrial mammals. However, before achieving this, the respective studies should be carried out by applying the selection criteria of prebiotics in aquatic organisms and analyzing the results with caution before applying them intensively and commercially.

Continued research will contribute to generating more options for prebiotic substances that can be applied to different areas, including aquaculture. However, it is vital to develop legislation that is as homogeneous as possible at the global level or in multinational regions, like the European Union’s on the application of prebiotics in aquaculture, which is known to have worked, to be able to apply them correctly and share knowledge among different research groups.

Global aquaculture has a growing demand that goes hand in hand with rapid population growth, so it would be interesting to consider the application of prebiotics, both those successfully tested and those potentially usable (e.g., agavin, chitosan, laminarian, and fucoidan), in species of fish with high global production and specific to each region with culture potential, in order to obtain more substantial production and higher quality of sustainable aquaculture products.

In Mexico, although studies have been carried out in recent years where prebiotics have been applied to fish of global importance, as well as in endemic fish, these and new prebiotics have yet to be tested with different doses and times of application; therefore, it is necessary to correctly define the optimal requirements to support the production of more profitable aquaculture.

## Figures and Tables

**Figure 1 animals-13-03607-f001:**
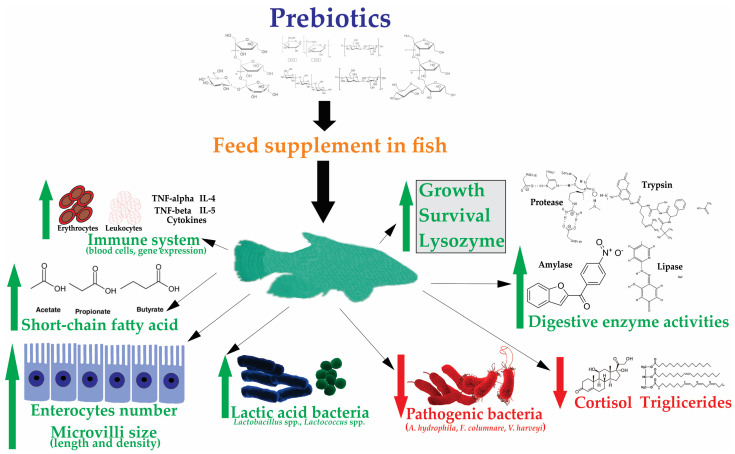
Action mechanisms of prebiotics identified in fish. Green arrows and letters indicate additive effects. Red arrows and letters indicate inhibitory effects. Black letters indicate the description of the action mechanisms.

**Figure 2 animals-13-03607-f002:**
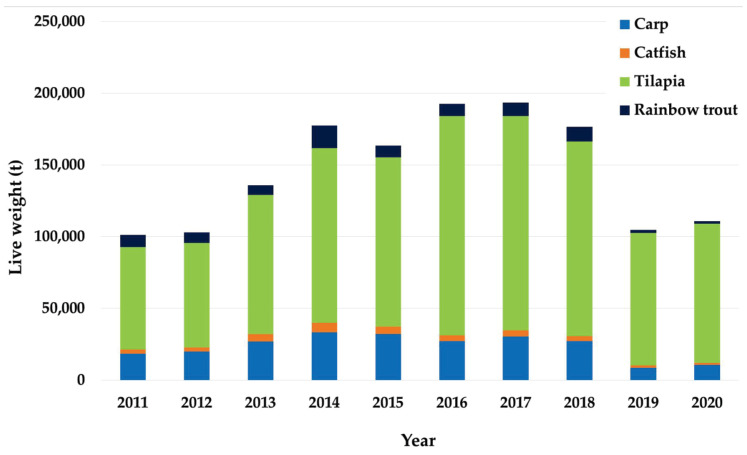
Aquaculture production (live weight in tons) of the main fish species cultivated in Mexico (2011–2020). (CONAPESCA 2021).

**Table 1 animals-13-03607-t001:** Summary of worldwide research on the effects of prebiotics supplements in fish aquaculture.

Prebiotics	Fish Species	Country (Experiment Conducted)	Dosage/ Application Time	Prebiotics Sources	Effects	References
Arabinoxylan-oligosaccharides-3-0.25, Arabinoxylan-oligosaccharides-32-0.30	*Acipenser baerii*	Belgium	2%/84 days	Wheat bran via extraction with endoxylanases	↑ Survival and phagocytic activity → Lysozyme activity ↑ Lactic acid bacteria and *Clostridium* sp. Arabinoxylan-oligosaccharides-32-0.30 ↑ growth Arabinoxylan-oligosaccharide-32-0.30 ↑ short-chain fatty acids (acetate and butyrate)	[29]
β-glucan	*Labeo* *rohita*	India	0.01, 0.025, 0.05%/56 days	*S. cerevisiae*	↑ Growth, phagocytosis, lysozyme, haemolytic complement, bactericidal activity	[30]
β-glucan	*Pseudosciaena crocea*	China	0.5 y 1%/56 days	*S. cerevisiae*	↑ Growth, lysozyme, phagocytosis, protection against *Vibrio harveyi*	[31]
β-glucan	*Oncorhynchus mykiss*	Iran	0.1 and 0.2%/60 days	*S. cerevisiae*	↑ Growth and lysozyme	[32]
β-glucan	*Oreochromis* *niloticus*	Thailand	0.1%/56 days	*S. cerevisiae*	↓ Bacteria *Aeromonas hydrophila* and *Flavobacterium columnare*	[33]
β-glucan	*Carassius auratus* var. *Pengze*	China	0.1%/70 days	*S. cerevisiae*	→ Growth, ↑ immune system, and microvilli size	[34]
β-glucan	*Trachinotus ovatus*	Vietnam	0.1, 0.2% and 0.4%/56 days	*S. cerevisiae*	↑ Growth in 0.1%	[35]
β-glucan	*Rutilus rutilus*	Poland	1%/14 days	*S. cerevisiae*	→ Growth ↑ Immune system	[36]
β-glucan	*Hyphessobrycon eques*	Brazil	0.05, 0.1 and 0.2%/42 days	*S. cerevisiae*	↑ Immune system in 0.2%	[37]
Cell wall (β-glucan and MOS)	*Sparus aurata*	Spain	0.1, 0.5 and 1%/28 days	*S. cerevisiae*	0.5 and 1% ↑ phagocytic activity 0.1% ↑ cytotoxic activity	[38]
Cell wall (β-glucan and MOS)	*Labeo* *rohita*	India	0.5%/15 days	*S. cerevisiae*	↑Phagocytic activity	[39]
Cell wall (β-glucan and MOS)	*O. niloticus*	Egypt	0.1 and 0.2%/60 days	*S. cerevisiae*	↑ Growth, white blood cell count ↑ Phagocytic activity and gene expression related to immune system in 0.2%	[40]
FOS	*Salmo salar* L.	Norway	0.1%/70 days	Biomar AS, Brande, Denmark	→ Growth	[41]
FOS, chitosan, MOS, β-glucan, XOS	Hybrid *Epinephelus lanceolatus* x *Epinephelus fuscoguttatus*	China	0.2, 0.5, 0.2, 0.1, 0.05%/28 days	Shandong Shengyuan Biotechnology Co., Ltd., Qingdao, China	MOS and XOS ↑ growth, survival of XOS ↑ protein in muscle	[42]
GOS	*Cyprinus carpio*	Poland	2%/50 days	Bi2tos^®^, Clasado Biosciences Ltd., Jersey, UK	↑ Immune system	[43]
Immunogen^®^ provided by Soroush Radian Co., Ltd., Tehran, Iran	*C. carpio*	Iran	0.05, 0.1, 0.15, and 0.25%/56 days	Immunogen^®^ provided by Soroush Radian Co., Ltd., Tehran, Iran	→ Growth Immunogen^®^ 0.15 y 0.25%, ↑ leucocytes	[44]
IMO	*Clarias gariepinus*	Malaysia	0.5%/56 days	Composed by combination of isomaltose, isomaltotriose, maltose, panose, maltotriose, glucose and others (24.5, 12.0, 6.1, 1.6, 1.5, 0.3, and 43.7%)	→ Growth	[45]
Inulin	*Salvelinus alpinus*	Norway	15%/28 days	Not available	↓ Enterocytes of hindgut	[46]
Inulin, Oligofructose (type of FOS), Lactosucrose	*Psetta maxima*	France	2%/26 days	Inulin obtained from chicory *Cichorium intybus*. Oligofructose produced via partial enzymatic hydrolysis of chicory Inulin. Lactosucrose obtained from the Ensuiko Sugar Refining Co. (Yokohama, Japan)	→ Survival of oligofructose, ↑ growth of inulin and lactosucrose, ↑ gut microbiota	[47]
Inulin	*Huso huso*	Iran	1, 2 and 3%/56 days	Chicory *Cichorium intybus*	↓ Growth and survival, ↓ Total bacteria, ↑ lactic acid bacteria	[48]
Inulin	*O. niloticus*	Egypt	0.5%/60 days	Chicory *Cichorium intybus*	↑ Growth and survival	[49]
Inulin, FOS	*O. mykiss*	Spain	0.5 and 1%/49 days	Inulin obtained from chicory *Cichorium intybus* roots (PREBIOFEED 88; Qualivet, Las Rozas, Spain). Fructooligosaccharides obtained viq partial enzymatic hydrolysis of inulin (Oligofructose from Beneo P95; Beneo-Orafti España SL, Barcelona, Spain)	↑ Growth, → gut bacteria (*Aeromonas* spp., *Pseudomonas* spp. and Gram-positive bacteria)	[50]
Inulin	*C. carpio*	Iran	0.5 and 1%/49 days	Provided by Orafti (Raffinerie Tirlemontoise, Tienen, Belgium)	→ Growth and enzymatic activity (lipase, protease, and amylase) ↑ Survival	[51]
Inulin, MOS	*Ctenopharyngodon idella*	China	0.2 and 2%/56 days	Inulin from chicory *Cichorium intybus* (Sigma, Saint Louis, MO, USA). MOS from yeast *Saccharomyces cerevisiae* (Fubon, Yichang, China)	Inulin and MOS 2% ↑ growth and bactericidal activity	[52]
Inulin	*O. niloticus*	Thailand	0.25 and 0.5%, 0.5 and 1%/56 days	Inulin from chicory *Cichorium intybus* (PREBIOFEED 88; Warcoing, Belgium)	↑ Growth, blood cell number, and lysozyme activity → Survival	[53]
Inulin	*O. mykiss*	Turkey	1 and 2%/56 days	Chicory roots *Cichorium intybus*	↑ Growth and survival in 1%, ↑ Digestive enzyme activities in 1%	[54]
Inulin	*O. niloticus*	Egypt	0.25, 0.5 and 1%/90 days	Chicory roots *Cichorium intybus*	↑ Growth in 0.25%	[55]
Inulin	*Pseudoplatystoma reticulatum*	Brazil	0.7%/12 days	Chicory roots *Cichorium intybus*	→ Growthand ↓ microvilli size	[56]
Inulin	*O. mykiss*	Iran	1, 2 and 3%/60 days	Inulin Orafti^®^ GR (Beneo Company, Tienen, Belgium)	↑ Growth and lysozyme activity	[57]
Inulin	*Pelteobagrus fulvidraco*	China	0.4%/70 days	Inulin Orafti^®^ GR (Beneo Company, Tienen, Belgium)	↑ Growth and butyric acid	[58]
MOS	*Dicentrarchus labrax*	Spain	0.2 and 0.4%/63 days	*S. cerevisiae*	↑ Growth	[59]
MOS	*O. mykiss*	Turkey	0.15, 0.3 and 0.45%/90 days	MOS were derived from the outer cell wall of the yeast *S. cerevisiae*	↑ Growth in MOS 0.15% ↑ Microvilli length in MOS 0.15 and 0.3%	[60]
MOS, FOS, GOS	*S. salar*	Norway	1%/120 days	MOS (Bio-Mos, Alltech Inc., Nicholasville, KY, USA), FOS from inulin (Encore Technologies, Plymouth, MN, USA), GOS (Friesland Foods Domo, Zwolle, The Netherland)	→ Growth and survival MOS 1% ↓ Lysozyme activity	[61]
MOS	*O. mykiss*	United Kingdom	0.2%/58 days	MOS (Bio-Mos, Alltech Inc., Lexington, KY, USA)were derived from the outer cell wall of the yeast *S. cerevisiae* strain 1026	↑ Microvilli length and density ↓ Gut bacteria *Aeromonas* spp. and *Vibrio* spp.	[62]
MOS	*D. labrax*	Egypt	0.1, 0.2, 0.3 and 0.4%/75 days	*S. cerevisiae*	↑ Growth y microvilli size ↑ Survival (0.1%)	[63]
MOS	*Pangasianodon hypophthalmus*	Malaysia	0.2, 0.4, 0.6 and 0.8%/84 days	*S. cerevisiae*	↑ Survival and lysozyme activity against pathogen *A. hydrophila*	[64]
MOS	Hybrid *E. lanceolatus* ♂ and *E. fuscoguttatus* ♀	China	0.3, 0.6, 1.0 and 2.0%/63 days	*S. cerevisiae*	↑ Lysozyme activity and microvilli length	[65]
Oligofructose	*Oreochromis* spp.	Thailand	0.5, 1.0%/28 days	Jerusalem Artichoke *Helianthus tuberosus*	↑ Growth, immune system	[66]
XOS	*D. labrax*	Tunisia	0.5 and 1%/84 days	Corncob *Zea mays*	↑ Growth (0.5%) and survival against pathogen *A. hydrophila* (1%)	[67]
XOS	*O. mykiss*	China	0.25, 0.5, 0.75 and 1.0%/56 days	Henan Hebi Taixin Technology Co., Ltd., Zibo, China	↑ Growth (1%) ↑ Microvilli height ↑ Lipase and amylase activity	[68]

Symbols represent increase (↑), no effect (→), or decrease (↓) in the response parameter of the prebiotics relative to the control. ♂ (male); ♀ (female).

**Table 2 animals-13-03607-t002:** Global inland, coastal, and marine aquaculture production of the major fish families and species in 2022. Modified of FAO (2022).

Family	Common Name, Species	Production (Million Tons)	Percentage
Inland aquaculture			
Centrarchidae	Largemouth black bass, *Micropterus salmoides*	0.62	1.3
Cichlidae	Nile tilapia, *O. niloticus* Tilapias nei, *Oreochromis* spp.	4.41 1.07	9 2.2
Clariidae	Clarias catfishes, *Clarias* spp.	1.25	2.5
Cyprinidae	Grass carp, *Ctenopharyngodon idellus*	5.79	11.8
	Silver carp, *Hypophthalmichthys molitrix*	4.9	10
	Common carp, *C. carpio*	4.24	8.6
	Catla, *Catla catla*	3.54	7.2
	Bighead carp, *Hypophthalmichthys nobilis*	3.19	6.5
	*Carassius* spp.	2.74	5.6
	Roho labeo, *L. rohita*	2.48	5.1
	Wuchang bream, *Megalobrama amblycephala*	0.78	1.6
	Black carp, *Mylopharyngodon piceus*	0.7	1.4
Pangasiidae	Striped catfish, *P. hypophthalmus*	2.52	5.1
Salmonidae	Rainbow trout, *O. mykiss*	0.74	1.5
	Subtotal of 15 major species	38.97	79.3
	Subtotal other species	10.15	20.7
	Total	49.12	100
Coastal and marine aquaculture			
Carangidae	Pompano, *T. ovatus*	0.16	1.9
	Japanese amberjack, *Seriola quinqueradiata*	0.14	1.6
Chanidae	Milkfish, *Chanos chanos*	1.17	14
Cichlidae	Nile tilapia, *O. niloticus*	0.11	1.3
Lateolabracidae	Japanese seabass, *Lateolabrax japonicus*	0.2	2.4
Latidae	Barramundi(=Giant seaperch), *Lates calcarifer*	0.11	1.3
Moronidae	European seabass, *D. labrax*	0.24	2.9
Mugilidae	Mullets nei, Mugilidae	0.29	3.5
Salmonidae	Atlantic salmon, *S. salar*	2.72	32.6
	Coho(=Silver) salmon, *Oncorhynchus kisutch*	0.22	2.7
	Rainbow trout, *O. mykiss*	0.22	2.6
Sciaenidae	Red drum, *Sciaenops ocellatus*	0.08	1
	Large yellow croaker, *Larimichthys croceus*	0.25	3
Serranidae	Groupers nei, *Epinephelus* spp.	0.23	2.7
Sparidae	Gilthead seabream, *S. aurata*	0.28	3.4
	Subtotal of 15 major species	6.42	77
	Subtotal other species	1.92	23
	Total	8.34	100

## Data Availability

The data presented in this study are available on request from the corresponding author.

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
