# Peer review of "Prebiotics in Global and Mexican Fish Aquaculture: A Review"

_animals, 2023, doi:10.3390/ani13233607_

Round 1

Reviewer 1 Report

Comments and Suggestions for Authors

Dear Authors, Although it most likely will be a great paper the inconsistency of the terminology prevented me from reviewing the whole paper. 

Comments on the Quality of English Language
Line 21-23: rephrase

Aquaculture is a better term for ‘producing commercial fish’ as the fishing for wild fish (the opposite of aquaculture) is called ‘commercial fishing’. Please think of replacing this throughout the manuscript

Line 26 What meant with fish of natural distribution? Wild fish? If that is the case please remove as you will never have probiotics in those. If not wild fish please explain more or use different phrase. Reading further I think you mean ‘endemic species’

Line 28: explain what is meant by fisheries. Normally this is referred to as wild caught fish activities. Not appropriate in this contecxt if so and should be removed

Line 30:  remove ‘fish’

Line 32: is "a substrate that is selectively utilized by host microorganisms conferring a health benefit” not a better definition of a prebiotic? lots of digestible compounds have a benefit to gut flora and in turn to our health.

Line 35: replace survive with boost. You can add increase survivability if that is applicable

Line 38: ‘gives a review’

Line 40: remove alternative as you are not suggesting one and offering alternatives to that single one.

Line 41 Mexico, as well as freshwater

Line 48. Word Fisheries used firstly used referring to wild caught fish and then, in the same sentence as an overarching term for all fish produced for consumption. First one should be betted explained

Reviewer 2 Report

Comments and Suggestions for Authors

This review paper is comprehensive and well-written. It provides good summary of use of prebiotics in global and Mexican fish aquaculture. This information is useful for world’s aquaculture industry. Some suggestions for revision:

1.      As a review paper, I do think a table of contents is necessary, which will guide better for the readers.

2.      A figure explaining the action mechanisms of prebiotic can be added. There is not a figure in this paper.

3.      The action mechanisms of prebiotic, especially those specific to fish species, and farming mode, should be introduced.

4.      The factors which influence the efficacy of prebiotics, for example the microbiota of fish intestine and the microbiota of farming environment should be discussed.

5.      Commercial use of prebiotics in global or Mexican market should be introduced. 

Reviewer 3 Report

Comments and Suggestions for Authors

My comments can be found in the attached file

Comments on the Quality of English Language

Minor editing of the English language required
